# Effectiveness, Immunogenicity and Harms of Additional SARS-CoV-2 Vaccine Doses in Kidney Transplant Recipients: A Systematic Review

**DOI:** 10.3390/vaccines11040863

**Published:** 2023-04-18

**Authors:** Renate Ilona Hausinger, Quirin Bachmann, Timotius Crone-Rawe, Nora Hannane, Ina Monsef, Bernhard Haller, Uwe Heemann, Nicole Skoetz, Nina Kreuzberger, Christoph Schmaderer

**Affiliations:** 1Department of Nephrology, Klinikum Rechts der Isar, TUM School of Medicine, Technical University of Munich, 81675 Munich, Germany; 2Evidence-Based Medicine, Department I of Internal Medicine, Faculty of Medicine and University Hospital Cologne, University of Cologne, 50937 Cologne, Germany; 3Institute for AI and Informatics in Medicine, Klinikum Rechts der Isar, TUM School of Medicine, Technical University of Munich, 81675 Munich, Germany

**Keywords:** SARS-CoV-2 vaccines, kidney transplant recipients, COVID-19, acute graft rejection

## Abstract

Background: Kidney transplant recipients (KTRs) who have a highly impaired immune response are in need of intensified and safe vaccination strategies to achieve seroconversion and prevent severe disease. Methods: We searched the Web of Science Core Collection, the Cochrane COVID-19 Study Register and the WHO COVID-19 global literature on coronavirus disease from January 2020 to 22 July 2022 for prospective studies that assessed immunogenicity and efficacy after three or more SARS-CoV-2 vaccine doses. Results: In 37 studies on 3429 patients, de novo seroconversion after three and four vaccine doses ranged from 32 to 60% and 25 to 37%. Variant-specific neutralization was 59 to 70% for Delta and 12 to 52% for Omicron. Severe disease after infection was rarely reported but all concerned KTRs lacked immune responses after vaccination. Studies investigating the clinical course of COVID-19 found remarkably higher rates of severe disease than in the general population. Serious adverse events and acute graft rejections were very rare. Substantial heterogeneity between the studies limited their comparability and summary. Conclusion: Additional SARS-CoV-2 vaccine doses are potent and safe in general terms as well as regarding transplant-specific outcomes whilst the Omicron wave remains a significant threat to KTRs without adequate immune responses.

## 1. Introduction

Vaccination remains the most effective tool to prevent severe disease after infection with SARS-CoV-2. The currently dominating variant of concern (VOC), Omicron (B.1.1.529), and its sublineages have shown to be less dangerous than the Delta strain for the general population with regard to severe disease [1,2]. Nevertheless, they continue to gain higher transmissibility and evasiveness to antibodies induced by previous infection or vaccination. Other preventive measures such as monoclonal antibodies have been developed to complete prevention strategies in seronegative patients but can hardly keep up with the high mutation rate of the virus [3,4]. For example, in the most recent preexposure prophylaxis trials, cilgavimab–tixagevimab in various dosages has been shown to be less effective with regard to the Omicron BA.4 strain, and its neutralization capacity against the Omicron BA.5 waned earlier than for other Omicron variants [5,6].

Kidney transplant recipients (KTRs) compose the largest group of people receiving an organ transplant [7] and belong to the most vulnerable patients regarding severe disease and COVID-19-related complications, due to their reduced kidney function, comorbidities and substantial immunosuppression.

Whilst pooled evidence exists for humoral and cellular responses to additional SARS-CoV-2 vaccine doses in solid organ transplant recipients, these results are not transferable to KTRs, who tend to be more immunocompromised than, for example, liver transplant recipients [8]. Various approaches such as additional vaccine doses, different vaccination strategies or a temporary reduction of immunosuppressive therapy have been under investigation. This systematic review of prospective studies aimed to evaluate the effectiveness of additional SARS-CoV-2 vaccine doses regarding cellular, humoral and variant-specific immunity and the incidence and severity of breakthrough infections, as well as general and transplant-specific safety in KTRs.

## 2. Methods

This study was registered with PROSPERO (CRD42022315409) and is accessible at Open Science Framework under “https://osf.io/nsyq4/ (accessed on 15 April 2023)”. The reporting of this systematic review follows the PRISMA checklist [9].

### 2.1. Data Sources and Searches

We searched the Web of Science Core Collection, the Cochrane COVID-19 Study Register and the WHO COVID-19 global literature on coronavirus disease up to 22 July 2022. The search strategies can be found in the Appendix A. Language restrictions were not applied.

### 2.2. Study Selection

We included randomized controlled studies or prospective observational studies reporting on at least 20 adult KTRs receiving a minimum of 3 SARS-CoV-2 vaccine doses with either vaccines authorized for use by the European Medicines Agency (EMA) or vaccines granted Emergency Use Listing by the World Health Organization (WHO). As of 22 February 2022 those were [10]:mRNA vaccines: BNT162b2 by BioNTech/ Pfizer (Mainz, Germany) mRNA-1273 by Moderna (Cambridge, MA, United States of America);Non-replicating viral vector vaccines: ChAdOx1 by Oxford/AstraZeneca (Cambride, United Kingdom), Ad26.COV2.S by Johnson & Johnson (New Jersey, United States of America);Inactivated viral vaccines: Sinovac by Sinovac Biotech (Beijing, China), BBIBP-CorV by Sinopharm (Beijing, China), BBV152 by Bharat Biotech (Hyderabad, India);Protein subunit viral vaccines: NVX-CoV2373 by Novavax (Gaithersburg, MA, United States of America);The non-replicating viral vector vaccine rAd5 by Gamaleya (Moscow, Russia) despite its pending approval due to its wide distribution.

Studies were considered eligible for inclusion if they reported one or more of the following predefined main outcomes: SARS-CoV-2- infection, the composite endpoint hospitalization or death, seroconversion, antibody titers and/or t-cell responses. Whenever available, we extracted the following predefined additional outcomes: Serious adverse events (SAE) and the development of de novo donor-specific antibodies (DSAs) as markers of alloimmunization, as well as the incidence of acute graft rejection after COVID-19 vaccination. Furthermore, we assessed variant-specific neutralization capacity against the delta and omicron VOC.

Five authors selected the studies on a web-based online platform (Rayyan [11]); each study being assessed independently by two authors in a two-step approach. Conflicts were resolved by group discussion involving a third author.

### 2.3. Data Extraction and Risk of Bias Assessment

Study and population characteristics, as well as intervention-related information, were extracted by two authors independently. We did not extract data on healthy controls; they were scarce and not matched to our cohort of KTRs regarding age or clinical condition.

The risk of bias in each study was independently assessed on the outcome level by two review authors and discussed until a consensus was reached. For observational studies without a comparison, a tool for overall prognosis studies currently under development was used (RoB-OPS [12]). The tool uses the following four domains: “participants”, “outcomes”, “analysis” and “reporting bias”. For the risk of bias in non-randomized trials comparing interventions, we used the ROBINS-I tool [13] and for randomized controlled trials (RCT) we used the RoB2 tool [14].

### 2.4. Data Synthesis and Analysis

There was substantial clinical heterogeneity; some studies reported immunologic outcomes only for participants with negative serostatus, co-interventions or relevant discrepancies in immunosuppressive therapy. Methodological diversity existed in terms of variances in study design, outcome definition and measurement. Thus, we did not pool the outcome rates into a meta-analysis but we present the data descriptively in forest plots without a combined effect estimate, or in table format. To explore the risk of bias visually, we excluded all studies with high risk of bias as predefined in our study protocol (Appendix A). For the visualization of results, we used RStudio (R: version 4.2.1; Rstudio: version 2022.07.1; package: meta, version 5.2-0) [15,16]. More information on the algorithms used can be found in the last chapter of the Appendix A.

For our prioritized outcomes, we assessed the certainty in the evidence using the Grading of Recommendations Assessment, Development, and Evaluation (GRADE) approach [17], starting at a high certainty of evidence according to the guidelines for overall prognosis studies [18]. The quality of evidence of each outcome was assessed using the four domains “Risk of bias”, “Inconsistency”, “Indirectness” and “Imprecision”. Certainty of evidence was rated according to limitations in these domains to very low, low, moderate or high certainty of evidence. The summary of findings tables were produced with the GRADEpro GDT software [19].

## 3. Results

### 3.1. Results of the Search

The literature search resulted in 1318 records after deduplication. After title and abstract screening, we assessed 235 full texts for eligibility and included 37 published studies into the review. At the time of our update search, 23 studies were ongoing or completed without published data (Figure 1).

### 3.2. Description of Studies

From the included studies, 26 were from Europe, 6 from the USA, 3 from Israel and Canada and 1 from Thailand and Uruguay, each. We identified 2 RCTs, 8 non-randomized studies comparing interventions and 27 prognostic studies without a control arm. Twenty-six studies investigated a study population containing seropositive KTRs after their previous vaccination series [20,21,22,23,24,25,26,27,28,29,30,31,32,33,34,35,36,37,38,39,40,41,42,43,44,45,46]. Eight studies included only seronegative KTRs [47,48,49,50,51,52] and three studies included only KTRs with low or no previous seroconversion [53,54,55].

The included studies investigated five different vaccines (Cominarty^®^, Spikevax^®^, Vaxzevria^®^, COVID-19 Vaccine Janssen^®^ and Coronavac^®^). Twenty-two studies reported on mRNA vaccines. One examined triple vaccination with viral vector vaccines [33], six studies compared triple vaccination with an mRNA vaccine and heterologous (mRNA/viral vector) vaccination [23,27,41,44,47,48] and two studies compared mRNA-1273 with BNT162b2 [26,27,28,29,30,31,32,33,34,35,36,37,38,39,40,41,42,43,44,45,46,47,48,49,50,51,52,53]. Only two of these nine studies were prospectively planned comparisons. Three studies exclusively investigated co-interventions or specific immunosuppressive regimens and their outcomes are reported separately. Of the remaining 34 studies, 19 reported transplant vintage, which was 6.7 years on average. More than 60% of KTRs received triple immunosuppressive therapy in 20 studies and less than 40% in 4 studies whilst no information was given in 10 studies. Thirty-three studies investigated immunogenicity parameters. Fourteen studies evaluated vaccine effectiveness reporting clinical outcomes after COVID-19 infection. Sixteen studies investigated safety outcomes including SAE, the development of de novo DSAs or acute graft rejections. The summary of findings table gives an overview of all investigated outcomes (Table 1). A summary of study characteristics can be found in Appendix A.

### 3.3. Seroconversion

We identified 29 studies including 2834 KTRs with reports on seroconversion (Figure 2a). De novo seroconversion ranged from 32 to 60% in 1515 KTRs after the 3rd and from 25 to 36% in 219 patients after the 4th vaccine dose (21 studies).

Studies reporting anti-S-IgG showed a tendency towards higher seroconversion rates than studies reporting on anti-RBD (Figure 2c). Two of the three studies that used both antibody assays achieved significantly higher results with the anti-S-IgG assay [20,29,32].

Primary vaccination with Coronavac^®^ yielded low response rates ranging from 32% for the vaccine booster with ChAdOx1 to 40% for the mRNA booster. No advantage was seen for one vaccination strategy over the other. Nineteen studies reported seroconversion in their whole transplanted cohort, regardless of previous humoral immune response, from here referred to as total seroconversion. The total seroconversion rate after three vaccine doses ranged between 39 and 92% for 1686 KTRs, being lowest for the triple vector vaccine regimen [33] and highest for a study by Tauzin, 2022, reporting on 30 KTRs who were more recently transplanted (median transplant vintage 24.48 months, IQR −4.8–275.16) and considerably younger (median age 46 years, IQR 23–73) than in the other studies and had partly received their previous vaccine doses before kidney transplantation (Figure 2b). Without these studies, total seroconversion was 53 to 77%. After four vaccine doses, total seroconversion ranged between 45 and 81% in 538 KTRs. Importantly, Roch, 2022, and Midtvedt, 2022, included only KTRs with low or no seroconversion after their primary vaccination series, whilst the other studies did not set limitations on study population and thus yielded higher rates of 72 to 81%. Three studies reported stable or increasing rates of seroconversion after an additional follow-up of three months [23,52,56] whereas two studies [23,29] found decreasing or stable seroconversion rates after six months.

One RCT and four non-randomized studies of interventions compared de novo seroconversion with homologous vs. heterologous vaccine regimens in 345 previously seronegative KTRs (Figure 2c). Overall, most estimates were imprecise although indicated a slight tendency towards more events in the heterologous vaccine regimens. In three non-randomized studies of interventions in 715 KTRs which compared total seroconversion with homologous vs. heterologous vaccine regimens (Figure 2c), no superiority could be shown for either vaccine strategy.

### 3.4. Other Correlates of Protection

Variant-specific total neutralization was 58 to 70% against the Delta VOC after three and four vaccine doses for 171 KTRs (Figure 2d). In contrast, four studies yielded considerably lower neutralization rates against the Omicron VOC ranging from 11 to 52% for 142 patients. De novo neutralization rates were 35 to 36% for the Delta and 5 to 12% for the Omicron VOC after three vaccine doses (Appendix A).

Due to the high variance in reporting units of antibody titers and assays directed against different spike domains, we refrained from direct comparisons and reported antibody titers before and after the additional vaccine dose descriptively in Appendix A.

Six studies reported t-cell responses after three or four vaccine doses: three studies measured t-cell responses with ELISpot, two used an interferon gamma release assay (IGRA) and one used an activation-induced marker assay (AIM) (Figure 2e, Appendix A). ELISpot analyses yielded much higher results than IGRA, limiting comparability. Three studies examined t-cell responses in a subset of their original cohort; their sample sizes are small and therefore possibly not representative of their population.

### 3.5. SARS-CoV-2 Infection

We identified 14 studies reporting on breakthrough infections after three or four vaccine doses and sorted the studies according to the study period to show vaccine efficacy for the individual SARS-CoV-2-waves (Figure 3c). In 659 KTRs the event rate ranged from 0 to 5% until the end of 2021 and in 891 KTRs from 5 to 25% when Omicron had become the dominant variant. In nine studies reporting serological characteristics of the infected KTRs the majority of them had no adequate seroresponse. KTRs were followed up for up to six months in five studies, up to three months in five studies and up to one month in four studies.

Four hospital admissions (three ICU admissions) or deaths occurred in 564 patients. Remarkably, none of these patients had seroconverted after vaccination.

We found two studies reporting exclusively on the clinical course of COVID-19 in KTRs during the Omicron wave. Malahe, 2022, reported on 45 infected KTRs (38 triple vaccinated) with 9 (6 triple vaccinated) patients in need of hospital admission of whom five were seronegative. In Wong, 2022, 7 out of 41 KTRs (23 had 3 vaccine doses) were admitted to ICU, 16 needed ward-based care and 1 patient died (triple vaccinated). These findings suggest that due to short follow-up in most studies reporting breakthrough infections, the rate of severe disease might not have been captured adequately.

### 3.6. Serious Adverse Events

Nine studies reported on SAE in 996 patients; eight studies reported no SAE. All five events that were possibly treatment-related were reported in BOOST-Tx 2021, without any difference according to vaccine type (homologous mRNA vs. heterologous). Seija, 2022, found recurrence of IgA-nephritis in one KTR after the 4th vaccine dose.

### 3.7. Development of Transplant-Specific Adverse Events

There were no reports of a decline in renal function after vaccination in general. However, KTRs have frequent assessments of creatinine levels in their routine visits, so even if data on renal function are not published, it can be assumed that a vaccination-associated decline in kidney function would have been presented. De novo DSAs were reported in 3 of 157 patients with an event rate from 0 to 0.5% (Figure 3d). In one study, the KTR who developed de novo DSAs had not adhered to immunosuppressive therapy [22]. However, the low information size limits our certainty in this outcome. Acute graft rejection after three or four vaccine doses was found in 0 to 0.2% in 10 studies on 981 KTRs. The two events that occurred were reported in KTRs receiving four vaccine doses (primary series with two doses of Coronavac and two booster doses of BNT162b2 each 30 days apart).

### 3.8. Co-Interventions and Subgroups

Four trials reported on seroconversion rate and safety investigating the reduction or hold of mycophenolic acid (MPA) and one study investigated changes to MPA-free regimens (Appendix A). Two of these four studies suggested a potential benefit regarding seroconversion for the hold or reduction of MPA, after MPA hold for four to five weeks. Three studies monitored the appearance of de novo DSAs; no events occurred. Importantly, all of these studies were small scale and too heterogenous to allow comparability.

Three studies reported only on KTRs with three vaccine doses treated with Belatacept. De novo seroconversion rates were severely impaired at 12 to 20% (Appendix A).

Patients who did not seroconvert had a lower median transplant age than those who seroconverted (Appendix A).

There was not enough information on immunosuppression, age, graft type, diabetic status and transplantation number to compare outcomes between subgroups.

### 3.9. Risk of Bias

Regarding immunogenicity outcomes, a risk of bias occurred due to missing data or in outcome measurement. In total, 4 out of 14 and 5 out of 15 studies reporting de novo and total seroconversion after three vaccine doses were at a high risk of bias. From studies of four vaccine doses, two out of four and one out of four studies were at a high risk of bias, respectively. One of four studies investigating variant-specific neutralization after three vaccine doses was at a high risk of bias and none after four vaccine doses. One of six studies investigating t-cell responses was at a high risk of bias.

From the 14 studies investigating COVID-19 disease after vaccination, 9 studies failed to adequately define the outcome or did not report how infection was assessed. None of these 14 studies performed routine SARS-CoV-2 testing and, thus, the risk of overseen infections is high.

Five out of nine studies failed to assess SAE systematically or reported the occurrence of SAE only narratively, without a definition or details on the outcome measurement.

Acute graft rejection was examined by nine studies. Four of these nine studies did not report on outcome measurement and were therefore rated as at a high risk of bias in this domain. None of the three studies reporting on the development of donor-specific antibodies was at a high risk of bias.

The risk of bias of each study is depicted in detail in Appendix A.

## 4. Discussion

This systematic review aimed to gather evidence on the immunogenicity, efficacy and safety of additional SARS-CoV-2 vaccine doses in KTRs. Our findings on seroconversion rates show that the probability of seroconversion decreases with the number of applied vaccine doses. However, up to 60 and 30% of previously negative KTRs seroconvert after the third and fourth vaccine doses and up to 80% of all vaccinated KTRs reach seroconversion after three and four vaccine doses in general.

This may not hold true for KTRs with a primary vaccine series with Coronavac^®^, for which lower immune responses have previously been reported [58,59]. More recent transplantation was a risk factor for non-seroconversion. Our data suggest non-inferior seroconversion in KTRs after the third heterologous vaccination as reported for other populations [60].

Nevertheless, the prevalence of neutralizing antibodies against the Omicron VOC after three vaccine doses was remarkably lower than in the general population, where 97% were able to neutralize the virus after three vaccine doses [61]. It follows that, as the presence of neutralizing antibodies correlates best with infection [62,63], KTRs need more vaccine doses for sufficient protection, even though seroconversion is present. Due to the high methodological variance in investigating t-cell responses the results were highly heterogenous. As cellular responses are generally robust in the healthy population, the impaired cellular response in KTRs is likely provoked by MPA and other immunosuppressives affecting lymphocytes and thus preventing the generation of a broad immune response [64,65,66].

Co-interventions such as the reduction of MPA peri-vaccination were insufficiently comparable to allow a conclusion. Larger, well-designed trials, for example CPAT-ISR, currently investigate this promising strategy to increase vaccine efficacy [67].

The rate of hospital admission and death in the studies that reported breakthrough infections in this review is low, which is explainable by the very short follow-up periods in most studies that reported primarily on immunogenicity outcomes. Two studies in our review reporting on the clinical course of Omicron infections in KTRs found alarming hospital admission rates between 16 and 50%. Further studies in KTRs [68,69] and other solid organ transplant recipients (SOTRs) support similar event rates, for example, a hospital admission rate of 16% was found for lung-transplanted patients in a nationwide German study [70,71]. As in our analysis the few participants with severe disease had low or absent immune responses, additional vaccine doses can provide efficient [72] protection from severe disease once a sufficient serologic response is reached. Recent studies confirm these findings [73,74].

The safety profile observed in clinical trials investigating three or more vaccine doses in the general population is consistent with our observations in KTRs [75,76,77,78].

Considering the development of de novo DSAs, our evidence is limited by the scarcity of information. Two events occurred, unrelated to the reduction of immunosuppression, in a study with 150 patients, but only two acute graft rejections were observed in a study with nearly one thousand patients [41]. This event rate is comparable to the occurrence of acute graft rejections reported in a systematic review and meta-analysis of the immunologic sequelae in SOTRs published before the COVID-19 pandemic [79]. One prospective observational study investigating the development of de novo DSAs in 150 KTRs and heart transplant recipients found that eight patients developed DSAs after their primary vaccine series but all of them had stable graft function [80].

A meta-analysis comparing rejections in SOTRs after SARS-CoV-2 vaccination vs. infection found six events post-vaccination and thirty events post-infection [81]. This underlines that the benefit of additional SARS-CoV-2 vaccine doses to prevent severe COVID-19 disease outweighs the risk of vaccine-induced graft rejection, with respect to the current variant.

Since new Omicron-specific vaccines contain the same mRNA platform with a few nucleotides exchanged [82], their safety profile is expected to be similar.

## 5. Limitations

Our study aimed to give a broad outline of important aspects of COVID-19 vaccination in KTRs rather than reporting on immunogenicity alone and is, to our knowledge, unique in this respect. However, it has several limitations. Firstly, the predefined outcomes relevant for inclusion in our systematic review focused on immunogenicity and breakthrough infections, which lowered the reporting on other outcomes, especially for the development of de novo DSAs. Furthermore, our confidence in the evidence of several outcomes is limited due to heterogeneity in study design and the lack of standardized assays used for outcome measurement. This relates particularly to the t-cell responses after vaccination, for which we cannot draw consistent conclusions due to the large number of different tests.

The investigation of predefined subgroups was not adequately possible, since information on immunosuppression regimens in context with positive or negative seroresponse hardly existed.

## 6. Future Directions

Our systematic review provides an evidence-based framework to guide clinical decision making, health care policies and future research in organ transplantation. Continued research is needed for alternative approaches to ensure the establishment of an effective immune response in KTRs, for example by the reduction of immunosuppression during vaccination or examination of the efficiency of the variant-adapted vaccines. Preexposure prophylaxis with monoclonal antibodies should still be considered an alternative worth investigating for seronegative KTRs.

## 7. Conclusions

Overall, our systematic review highlights the importance and effectiveness of three and four SARS-CoV-2 vaccine doses for the induction of humoral responses and prevention of severe disease. Nevertheless, protection against infection was impaired during the Omicron VOC. In KTRs vaccinated thrice, Omicron-specific neutralization was low and, thus, more vaccine doses are required to induce protective levels. The benevolent safety profile, which also applies to transplant-specific outcomes, permits this strategy.

## Figures and Tables

**Figure 1 vaccines-11-00863-f001:**
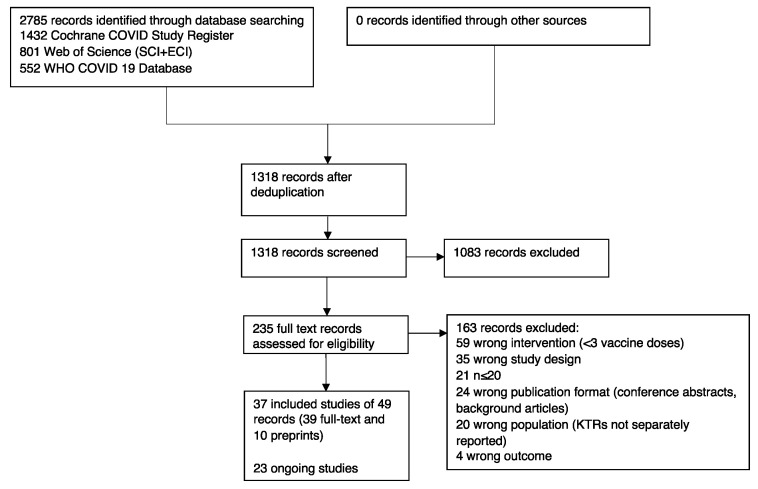
PRISMA flowchart: selection process for articles included in the systematic review.

**Figure 2 vaccines-11-00863-f002:**
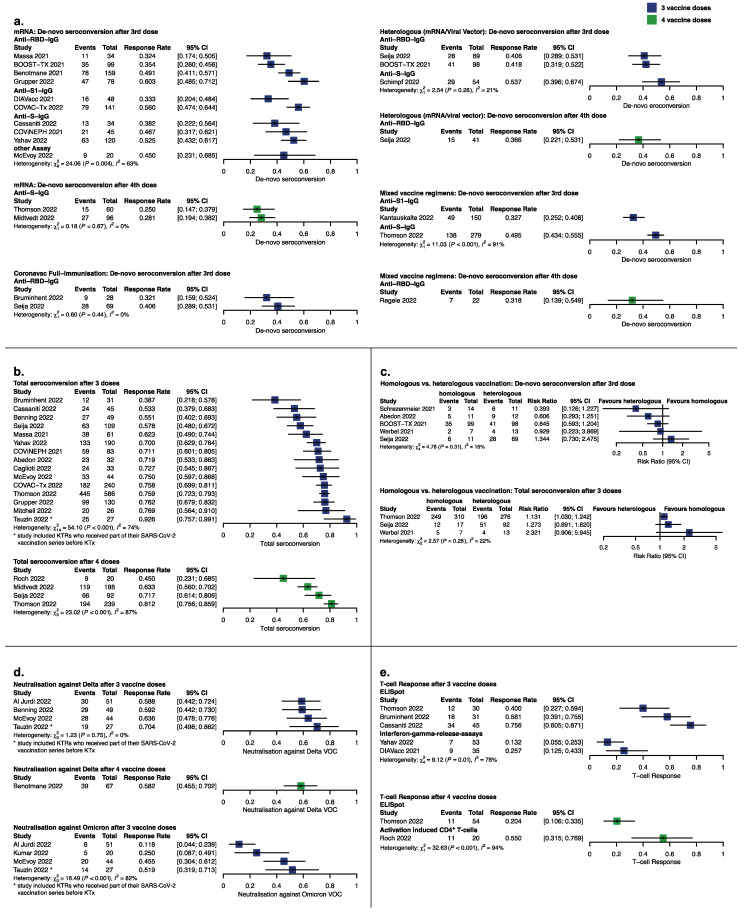
Seroconversion and seroresponse after SARS-CoV-2 vaccination depicted as forest plots. (**a**) De novo seroconversion after the third and fourth vaccine dose stratified according to vaccine type and assay used to define seroconversion. (**b**) Total seroconversion (KTRs seropositive after last vaccine dose including previously seroconverted KTRs) after three and four vaccine doses. (**c**) Studies comparing de novo and total seroconversion in homologous (mRNA) and heterologous (mRNA/viral vector) vaccination strategies. (**d**) Variant-specific neutralization after three and four vaccine doses. (**e**) T-cell responses stratified according to assay after three and four vaccine doses. [20,22,23,25,26,27,28,29,31,32,33,34,37,39,40,41,42,43,44,46,47,48,49,51,54,55,56,57].

**Figure 3 vaccines-11-00863-f003:**
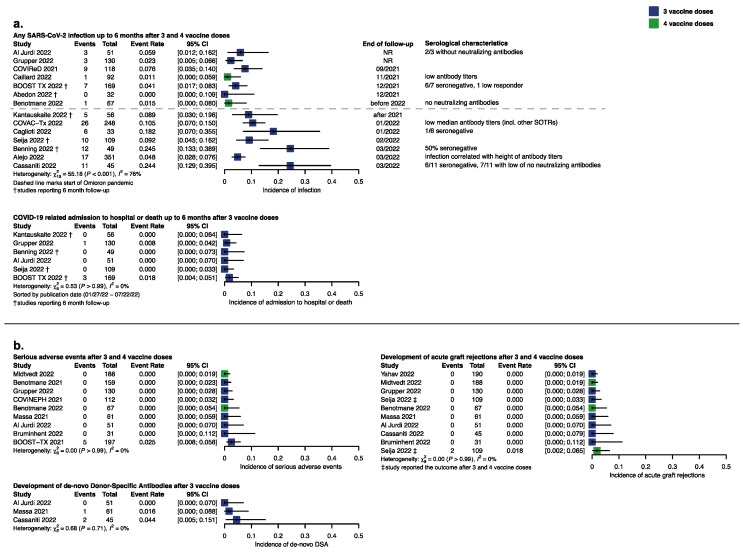
Response and event rate for prioritized efficacy and safety outcomes depicted as forest plots. (**a**) Incidence of SARS-CoV-2 breakthrough infection according to observation period with serological characteristics of infected KTRs and incidence of COVID-19-related hospital admission or death. (**b**) General and transplant-specific safety outcomes after 3 and 4 vaccine doses [20,21,23,26,30,32,33,34,37,41,42,43,48,49,53,55,56,57].

**Table 1 vaccines-11-00863-t001:** Summary of findings.

Outcome	No of Studies	Study Design	Effect			Certainty of Evidence
			No of Events	No of Individuals	Event Rate95 CI	
De novo Seroconversion up to 3 months after 3 vaccine doses	16	9 prospective observational studies, 6 NRSI, 1 RCT	686	1495	0.321 to 0.609	⨁⨁◯◯ Low ^a,b,j^
Total Seroconversion up to 3 months after 3 vaccine doses	15	4 NRSI, 11 prospective observational studies	1255	1777	0.387 to 0.926	⨁⨁◯◯ Low ^a,b,j^
De novo Seroconversion up to 3 months after 4 vaccine doses	4	2 NRSI, 2 prospective observational studies	64	219	0.25 to 0.366	⨁◯◯◯ Very low ^c,d,e,j^
Total Seroconversion up to 3 months after 4 vaccine doses	4	2 prospective observational studies	388	539	0.45 to 0.812	⨁⨁◯◯ Low ^a,c,j^
Neutralization against Omicron VOC after 3 vaccine doses	4	prospective observational studies	50	142	0.118 to 0.704	⨁◯◯◯ Very low ^c,f,e,j^
Neutralization against Delta VOC after 3 vaccine doses	4	prospective observational studies	101	171	0.51 to 0.63	⨁⨁◯◯ Low ^c,e,j^
Serious adverse events	9	1 RCT, 1, NRSI, 7 prospective observational studies	5	996	0 to 0.025	⨁⨁◯◯ Low ^g,i,j^
COVID-19-related hospitalization or death up to 6 months	6	1 RCT, 1NRSI, 4 prospective observational studies	4	564	0 to 0.018	⨁◯◯◯ Very low ^g,h,i,j^
Acute graft rejections	9	1 NRSI, 8 prospective observational studies	2	872	0 to 0.018	⨁⨁◯◯ Low ^g,i,j^
Development of de novo donor-specific antibodies	3	prospective observational studies	3	157	0 to 0.044	⨁⨁◯◯ Low ^c,i,j^

^a^ downgraded one level because of inconsistency due to serious heterogeneity, ^b^ downgraded one level because of imprecision due to wide confidence intervals, ^c^ downgraded one level because of risk of bias due to low number of studies ^d^ downgraded one level because of risk of bias due to confounding, ^e^ downgraded one level because of imprecision due to small information size and wide confidence intervals in most studies. ^f^ downgraded two levels because of very serious heterogeneity ^g^ downgraded one level because of risk of bias due to limitations in outcome definition and measurement, ^h^ downgraded one level because of risk of bias due to missing information on selection bias, ^i^ downgraded one level due to small information size, ^j^ Started with high level of evidence according to guidelines for overall prognosis studies. Certainty of evidence: ⨁◯◯◯ very low certainty of evidence, ⨁⨁◯◯ low certainty of evidence.

## Data Availability

The datasets generated and analyzed during the current study are available from the corresponding author on reasonable request.

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
