# Peer review of "Effectiveness, Immunogenicity and Harms of Additional SARS-CoV-2 Vaccine Doses in Kidney Transplant Recipients: A Systematic Review"

_vaccines, 2023, doi:10.3390/vaccines11040863_

Round 1

Reviewer 1 Report

Authors have explored the need for intensified and safe vaccination strategies to achieve seroconversion and prevent SARS-CoV-2 severe disease in kidney transplant recipients (KTRs) with impaired immune responses. Authors searched the Web of Science Core Collection, the Cochrane Covid-19 Study Register, and the WHO COVID-19 Global Literature on Coronavirus Disease from January 2020 to July 2022 for prospective studies that assessed immunogenicity and efficacy after three or more SARS-CoV-2 vaccine doses. Authors have analysed 37 studies on 3429 patients; de-novo seroconversion after three and four vaccine doses ranged from 32 to 60% and 25 to 37%, respectively. They have observed variant-specific neutralisation of 59 to 70% in the Delta strain and 12 to 52% for the Omicron strain. Furthermore, the authors have concluded that additional SARS-CoV-2 vaccine doses are potent and safe in general terms as well as regarding transplant-specific outcomes, whilst the Omicron wave remains a significant threat to KTRs without an adequate immune response. The authors have compiled excellent data. Generally, the manuscript is well written. Below, I have provided numerous suggestions, please incorporate, if possible:

  • Authors may include data related to fluctuations in renal function after vaccination.
  • Authors may include quantification data of T cell responses after vaccination.
  • Please include cellular response rate data after severe acute respiratory syndrome coronavirus vaccination.
  • Are there any indications of elevated creatinine in biopsies after vaccination?

Author Response

April 5th, 2023

Dear Editor,

Dear Reviewers,

please find attached our revised manuscript "Effectiveness, Immunogenicity and Harms of Additional SARS-CoV-2 Vaccine Doses in Kidney Transplant Recipients: A Systematic Review”.

In the attached letter we answer the comments of the reviewers and we believe that the revision significantly improved the quality of the manuscript.

The other coauthors have contributed in drafting of the manuscript and revising it critically.

In the Word article file of the manuscript, the changes were indicated with track change as required by the MDPI Vaccines Editorial Office.

Renate Hausinger and Quirin Bachmann were equally involved in all steps of this project from planning up to the present revision including also equal number of extracted and analyzed studies and an agreement was formed upon start of the project to share first authorship. Thus, we kindly ask for your acceptance of equally contributed first authorship.

We hope that our manuscript can be considered now for publication in Vaccines.

Sincerely,

Renate Hausinger and Quirin Bachmann

Point-by-Point response:

We thank the Editor and the Reviewers for evaluating our manuscript. In addressing each question in detail, we hope to meet the Editor’s and Reviewers’ demands and requirements.

Below, please find the point-by-point review with questions/comments (bold), answers (italic) and the manuscript passages and/or figures respectively (regular, changes highlighted in yellow):

For better overview please see the attached pdf file including the point-by-point-response.

Authors may include data related to fluctuations in renal function after vaccination.

We thank Reviewer 1 for the valuable input and hope to have answered the following questions in sufficient extent.

Five of the 37 included studies reported on renal function before and after vaccination without fluctuations. Two of these 5 studies reported creatinine after vaccination, the other 3 studies only reported no alterations of graft function in the text. Most studies included at least baseline kidney function in their patient characteristics or reported narratively that there hadn’t been deterioration in kidney function without providing data. 10 studies reported on graft rejections, which usually are suspected when renal function begins to decline. Nonetheless, data on kidney function in general after vaccination was not reported. So it can be hypothesized that there were no fluctuations in kidney function.

Furthermore, KTRs have frequent assessments of creatinine levels in their routine visits, so even if data on renal function are not published, it can be assumed that a vaccination-associated decline in kidney function would have been published.

p.7, 229-232:

There were no reports of a decline in renal function after vaccination in general. However, KTRs have frequent assessments of creatinine levels in their routine visits, so even if data on renal function are not published, it can be assumed that a vaccination-associated decline in kidney function would have been presented.

Authors may include quantification data of T cell responses after vaccination.

We thank the reviewer for this advice. Quantification data on t-cell response would be highly interesting and relevant to establish a standardized range of values one would expect after certain vaccine doses.

Our attempt to collect these data has failed because of two reasons:

First, as listed in detail in Supp. Table 6, different methods have been used for determination of t-cell response (e.g. Interferon gamma release assays, ELISpot, FACS analysis). This already complicated comparability of KTRs with positive responses in either of the tests.

Second, only two of 7 available studies for t-cell response reported quantitative values additionally to the number of KTRs with t-cell response (BOOST-TX2021 and Cassaniti 2022) in two different assays. Reasonable summary of these data was not possible.

p.10, 301-306:

Due to the high methodological variance in investigating t-cell response results were highly heterogenous. As cellular responses are generally robust in the healthy population, the impaired cellular response in KTRs is likely provoked by MPA and other immunosuppressives affecting lymphocytes and thus preventing the generation of a broad immune response. 1-3

p.11, 348-349:

This relates particularly to the t-cell response after vaccination, for which we cannot draw consistent conclusions due to the large number of different tests.

Please include cellular response rate data after severe acute respiratory syndrome coronavirus vaccination.

We agree with the reviewer about the importance of cellular response after vaccination.

t-cell response rates after vaccination can be found in Figure 2e and in Supp. Table 6 with more details on the studies. As b-cell response was not a predefined endpoint in our protocol, we had not assessed it. However, searching the included studies for b-cell response yielded rather limited evidence: only one study reported that relative and absolute count of RBD-specific CD19+ b-cells did not change between the second and third vaccine dose but responders showed higher absolute numbers of RBD+ b-cells after vaccination (Schrezenmeier 2021). In general, publications on B-cell response and especially after additional vaccine doses are scarce.

Whereas t- and b-cell responses to the vaccine are generally robust in the healthy (Sahin 2021, https://doi.org/10.1038/s41586-021-03653-6, Widge 2021 https://doi.org/10.1056/nejmc2032195), deficient cellular response has been associated with immunosuppressive medication by several authors (Schrezenmeier 2021, Werbel 2023 https://doi.org/10.1016/j.ajt.2023.03.014) as for example the use of mycophenolate mofetil and calcineurininhibitors directly inhibits lymphocytes. Thus, impaired cellular response can offer another explanation for deficient vaccine response in KTRs.

Since data about b-cell response is so limited in the included studies we decided not to include it into the manuscript and hope for the reviewer’s consent.

Are there any indications of elevated creatinine in biopsies after vaccination?

As discussed above, precise information about changes in kidney function in the present literature is scarce and in most instances no information about graft function after vaccination is available.

Rise in creatinine (besides rise in proteinuria) usually is the reason for kidney biopsy and thus, predeceases diagnosis of graft rejection. Ten studies reported on graft rejection but only Seija 2022 reported 2 events in 109 cases (fig. 3e). Thus, we concluded that there is no general nor transplant specific risk in 3rd and 4th vaccinations against SARS-CoV-2.

p.7, 235-238:

Acute graft rejection after three or four vaccine doses was found in 0 to 0.2% in 10 studies on 981 KTRs. The two events that occurred were reported in KTRs receiving four vaccine doses (primary series with two doses Coronavac and two booster doses of BNT162b2 each 30 days apart).

p.11, 369-370:

The benevolent safety profile which also applies to transplant-specific outcomes permits this strategy.

Reviewer 2 Report

A well-designed and conducted study according to the latest 2020 PRISMA guidelines.

Only one comment: In the supplement, a few tables seem to be cut, thus editing is required for the supplement to be well presented.

Author Response

April 5th, 2023

Dear Editor,

Dear Reviewers,

please find attached our revised manuscript "Effectiveness, Immunogenicity and Harms of Additional SARS-CoV-2 Vaccine Doses in Kidney Transplant Recipients: A Systematic Review”.

In the attached letter we answer the comments of the reviewers and we believe that the revision significantly improved the quality of the manuscript.

The other coauthors have contributed in drafting of the manuscript and revising it critically.

In the Word article file of the manuscript, the changes were indicated with track change as required by the MDPI Vaccines Editorial Office.

Renate Hausinger and Quirin Bachmann were equally involved in all steps of this project from planning up to the present revision including also equal number of extracted and analyzed studies and an agreement was formed upon start of the project to share first authorship. Thus, we kindly ask for your acceptance of equally contributed first authorship.

We hope that our manuscript can be considered now for publication in Vaccines.

Sincerely,

Renate Hausinger and Quirin Bachmann

Point-by-Point response:

We thank the Editor and the Reviewers for evaluating our manuscript. In addressing each question in detail, we hope to meet the Editor’s and Reviewers’ demands and requirements.

Below, please find the point-by-point review with questions/comments (bold), answers (italic) and the manuscript passages and/or figures respectively (regular, changes highlighted in yellow):

For better overview please see the attached pdf file including the point-by-point-response and the revised Supplementary material.

Only one comment: In the supplement, a few tables seem to be cut, thus editing is required for the supplement to be well presented.

We are grateful for the positive comment and the acknowledgement of our methodological work.

We re-edited the supplementary table and double checked after downloading it from the website to ensure correct placement of all tables.

Reviewer 3 Report

This systematic review of prospective studies aimed to evaluate the effectiveness  of additional SARS-CoV-2 vaccine doses regarding cellular, humoral and variant-specific 51 immunity, the incidence and severity of breakthrough infections as well as general and  transplant-specific safety in KTRs. 

(PUT INSERTS IN A DIFFERENT COLOR FONT TO IDENTIFY CHANGES IN THE ARTICLE)

Include in article

1 - Adapt the title to the content of the article - Indicate that it is a systematic review.

2 - Abstract - remove the numbers

3 - Include a list of abbreviations before the abstract

4 - Include limitations after the discussion

5 - Include future directions after limitations

6 - Include the conclusion according to the findings at the end of the article

7 - The link to the supplemental material is not working

8- Improve discussion - Discuss with data from the most relevant articles today,

9 - Supplementary material

A -Include the database used to perform the analysis

B -Include the algorithms and scripts used in the software to perform the statistical analysis.

Author Response

April 5th, 2023

Dear Editor,

Dear Reviewers,

please find attached our revised manuscript "Effectiveness, Immunogenicity and Harms of Additional SARS-CoV-2 Vaccine Doses in Kidney Transplant Recipients: A Systematic Review”.

In the attached letter we answer the comments of the reviewers and we believe that the revision significantly improved the quality of the manuscript.

The other coauthors have contributed in drafting of the manuscript and revising it critically.

In the Word article file of the manuscript, the changes were indicated with track change as required by the MDPI Vaccines Editorial Office.

Renate Hausinger and Quirin Bachmann were equally involved in all steps of this project from planning up to the present revision including also equal number of extracted and analyzed studies and an agreement was formed upon start of the project to share first authorship. Thus, we kindly ask for your acceptance of equally contributed first authorship.

We hope that our manuscript can be considered now for publication in Vaccines.

Sincerely,

Renate Hausinger and Quirin Bachmann

Point-by-Point response:

We thank the Editor and the Reviewers for evaluating our manuscript. In addressing each question in detail, we hope to meet the Editor’s and Reviewers’ demands and requirements.

Below, please find the point-by-point review with questions/comments (bold), answers (italic) and the manuscript passages and/or figures respectively (regular, changes highlighted in yellow):

For better overview please see the attached pdf file including the point-by-point-response and the revised Supplementary material.

1 - Adapt the title to the content of the article - Indicate that it is a systematic review.

We thank the reviewer for the comment and changed the title to “Effectiveness, Immunogenicity and Harms of Additional SARS-CoV-2 Vaccine Doses in Kidney Transplant Recipients: A Systematic Review”.

2 - Abstract - remove the numbers

Numbers in the abstract were removed as suggested.

3 - Include a list of abbreviations before the abstract

We agree with the reviewer that this improves clarity for the reader and included a list of abbreviations in alphabetical order.

p.1, 17-22:

Abbreviations: AIM, activation induced marker assay; COVID-19, coronavirus disease 2019; DSA, donor-specific antibodies; EMA, European Medicines Agency; GRADE, grading of recommendations assessment, development, and evaluation; IGRA, interferon gamma release assay; KTR, kidney transplant recipient; MPA, mycophenolic acid; NRSI, non-randomized study of intervention; RBD, receptor binding domain; RCT, randomizedcontrolled trial; SAE, Serious adverse events; SARS-CoV-2, severe acute respiratory syndrome-coronavirus-2; SOTR, solid organ transplant recipient; VOC, variant of concern; WHO, World Health Organization.

4 - Include limitations after the discussion

We thank the reviewer for advice on structure of the manuscript. Limitations were placed and adjusted as extra title number 5.

p.11, 340-352:

  1. Limitations

Our study aimed to give a broad outline of important aspects of COVID-19 vaccination in KTRs rather than reporting on immunogenicity alone and is to our knowledge unique in this respect. However, it has several limitations. Firstly, the predefined out-comes relevant for inclusion in our systematic review focused on immunogenicity and breakthrough infections, which lowered the reporting on other outcomes, especially for development of de-novo DSA. Furthermore, our confidence in the evidence of several outcomes is limited due to heterogeneity in study design and the lack of standardized assays used for outcome measurement. This relates particularly to the t-cell response after vaccination, for which we cannot draw consistent conclusions due to the large number of different tests.

The investigation of predefined subgroups was not adequately possible, since infor-mation on immunosuppression regimens in context with positive or negative se-roresponse hardly existed. 

5 - Include future directions after limitations

Future directions were included as title number 6.

p.11, 355-362:

Our systematic review provides an evidence-based framework to guide clinical decision making, health care policies and future research in organ transplantation. Continued research is needed for alternative approaches to ensure the establishment of an effective immune response in KTRs, for example by the reduction of immunosuppression during vaccination or examination of the efficiency of the variant-adapted vaccines. Preexposure prophylaxis with monoclonal antibodies should still be considered an alternative worth investigating for seronegative KTRs.

6 - Include the conclusion according to the findings at the end of the article

We included the conclusion after “6. Future directions” and completed it according to the findings.

p.11, 358-365:

  1. Conclusion

Overall, our systematic review highlights the importance and effectiveness of three and four SARS-CoV-2 vaccine doses for the induction of humoral response and prevention of severe disease. Nevertheless, protection against infection was impaired during the Omicron VOC. In KTRs vaccinated thrice, Omicron-specific neutralization was low and thus, more vaccine doses are required to induce protective levels. The benevolent safety profile which also applies to transplant-specific outcomes permits this strategy.

7 - The link to the supplemental material is not working

Thank you for the comment. Currently the link seems to be working. If there is any further problem please contact us so we can identify the problem with the help of the editorial office.

8- Improve discussion - Discuss with data from the most relevant articles today,

We agree with the reviewer on the importance of an up-to-date discussion and included literature. The following changes were made to the introduction and discussion:

p.2, 48-51:

For example, in the most recent preexposure prophylaxis trials with cilgavimab-tixagevimab in various dosages has been shown to be less effective with regard to the Omicron BA.4 strain and its neutralization capacity against the Omicron BA.5 waned earlier than for other Omicron variants 4 5.

p.10, 301-306:

Due to the high methodological variance in investigating t-cell response results were highly heterogenous. As cellular responses are generally robust in the healthy population, the impaired cellular response in KTRs is likely provoked by MPA and other immunosuppressives affecting lymphocytes and thus preventing the generation of a broad immune response. 1-3

p.10, 308-310:

Larger, well-designed trials like CPAT-ISR currently investigate this promising strategy to increase vaccine efficacy 6.

p.11, 320-321:

Recent studies confirm these findings 7 8.

9 - Supplementary material

A -Include the database used to perform the analysis

We view the database extracted from the included literature as our intellectual property. Upon reasonable request we agree to share the database to ensure a maximum level of scientific transparency. Naturally, we will provide the database to be reviewed by the reviewers and editorial board if desired. As excel files are not supported by the submission platform it will have to be sent via email.

B -Include the algorithms and scripts used in the software to perform the statistical analysis.

We agree with the reviewer that detailed information about the statistic and methodology used to generate meta-data must be provided. We added explanation about the algorithms and scripts in the supplementary material (Supplementary material p.28) and referred to it in the main manuscript.

p.3, 118-120:

For visualization of results, we used RStudio (R: version 4.2.1, Rstudio: version 2022.07.1, package: meta, version 5.2-0)9 10. More information on the used algorithms can be found in the last chapter of the supplementary material.

Round 2

Reviewer 3 Report

The authors responded adequately to the comments and there was a significant improvement in the manuscript.